# THE INGREDIENTS OF REAL-WORLD ROBOTIC REINFORCEMENT LEARNING

**Henry Zhu**[*1]**, Justin Yu**[*1]**, Abhishek Gupta**[*1]**, Dhruv Shah**[1]**,**
**Kristian Hartikainen**[2]**, Avi Singh**[1]**, Vikash Kumar**[3]**, Sergey Levine**[1]
[1] University of California, Berkeley    [2] University of Oxford    [3] University of Washington

## ABSTRACT

The success of reinforcement learning for real world robotics has been, in many cases limited to instrumented laboratory scenarios, often requiring arduous human effort and oversight to enable continuous learning. In this work, we discuss the elements that are needed for a robotic learning system that can continually and autonomously improve with data collected in the real world. We propose a particular instantiation of such a system, using dexterous manipulation as our case study. Subsequently, we investigate a number of challenges that come up when learning without instrumentation. In such settings, learning must be feasible without manually designed resets, using only on-board perception, and without hand-engineered reward functions. We propose simple and scalable solutions to these challenges, and then demonstrate the efficacy of our proposed system on a set of dexterous robotic manipulation tasks, providing an in-depth analysis of the challenges associated with this learning paradigm. We demonstrate that our complete system can learn without any human intervention, acquiring a variety of vision-based skills with a real-world three-fingered hand. Results and videos can be found at `https://sites.google.com/view/realworld-rl/`.

## 1 INTRODUCTION

Reinforcement learning (RL) can in principle enable autonomous systems, such as robots, to acquire a large repertoire of skills automatically. Perhaps even more importantly, reinforcement learning can enable such systems to *continuously improve* the proficiency of their skills from experience. However, realizing this in reality has proven challenging: even with reinforcement learning methods that can acquire complex behaviors from high-dimensional low-level observations, such as images, the assumptions of the reinforcement learning problem setting do not fit cleanly into the constraints of the real world. For this reason, most successful robotic learning experiments have employed various kinds of environmental instrumentation in order to define reward functions, reset between trials, and obtain ground truth state (Levine et al., 2016; Haarnoja et al., 2018a; Kumar et al., 2016; Andrychowicz et al., 2018; Zhu et al., 2019; Chebotar et al., 2016; Nagabandi et al., 2019; Gupta et al., 2016). In order to practically and scalably deploy autonomous learning systems that improve continuously through real-world operation, we must lift these limitations and design algorithms that can learn under the constraints of real-world environments, as illustrated in Figure 2.

We propose that overcoming these challenges in a scalable way requires designing robotic systems that possess three capabilities: they are able to (1) learn from their own raw sensory inputs, (2) assign rewards to their own trials without hand-designed perception systems or instrumentation, and (3) learn continuously in non-episodic settings without requiring human intervention to manually reset the environment. A system with these capabilities can autonomously collect large amounts of real world data – typically crucial for effective generalization – without significant instrumentation in each training environment, an example of which is shown in Figure 1. If successful, this would lift a major constraint that stands between current reinforcement learning algorithms and the ability to learn scalable, generalizable, and robust real-world behaviors. Such a system would also bring us significantly closer to the goal of embodied learning-based systems that improve continuously through their own real-world experience.

---

*These authors contributed equally. Correspondence to `henryzhu@berkeley.edu`.

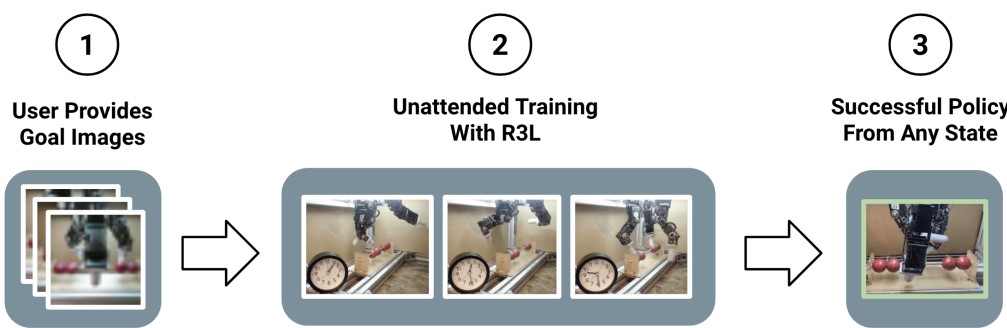

Figure 1: Illustration of our proposed instrumentation-free system requiring minimal human engineering. Human intervention is only required in the goal collection phase (1). The robot is left to train unattended (2) during the learning phase and can be evaluated from arbitrary initial states at the end of training (3). We show sample goal and intermediate images from the training process of a real hardware system

Having laid out these requirements, we propose a practical instantiation of such a learning system. While prior works have studied many of these issues in isolation, combining them into a complete real-world learning system presents a number of challenges, as we discuss in Section 3. We provide an empirical analysis of these issues, both in simulation and on a real-world robotic system, and propose a number of simple but effective solutions that together produce a complete robotic learning system. This system can autonomously learn from raw sensory inputs, learn reward functions from easily available supervision, and learn without manually designed reset mechanisms. We show that this system can learn dexterous robotic manipulation tasks in the real world, substantially outperforming ablations and prior work.

## 2 THE STRUCTURE OF A REAL-WORLD RL SYSTEM

The standard reinforcement learning paradigm assumes that the controlled system is represented as a Markov decision process with a state space $\mathcal{S}$, action space $\mathcal{A}$, unknown transition dynamics $\mathcal{T}$, unknown reward function $\mathcal{R}$, and a (typically) episodic initial state distribution $\rho$. The goal is to learn a policy that maximizes the expected sum of rewards via interactions with the environment.

Although this formalism is simple and concise, it does not capture all of the complexities of real-world robotic learning problems. If a robotic system is to learn continuously and autonomously in the real world, we must ensure that it can learn under the actual conditions that are imposed by the real world. To move from the idealized MDP formulation to the real world, we require a system that has the following properties. **First**, all of the information necessary for learning must be obtained from the robot's own sensors. This includes information about the state and necessitates that the policy must be learned from high-dimensional and low-level sensory observations, such as camera images. **Second**, the robot must also obtain the *reward signal* itself from its own sensor readings. This is exceptionally difficult for all but the simplest tasks (e.g., reward functions that depend on interactions with specific objects require perceiving those objects explicitly). **Third**, we must be able to learn without access to episodic resets. A setup with explicit resets quickly becomes impractical in open-world settings, due to the requirement for significant human engineering of the environment, or direct human intervention during learning. While this list may not exhaustively enumerate all the components necessary for an effective real-world learning system, we posit that the properties listed here are fundamental to building such systems.

While some of the components discussed above can be tackled in isolation by current algorithms, there are considerable challenges inherent to assembling all these components into a complete learning system for real world robotic learning. In the rest of this section, we outline the challenges associated with each component, then discuss the challenges associated with combining these components in Section 3, and then proceed to address these challenges in Section 4.

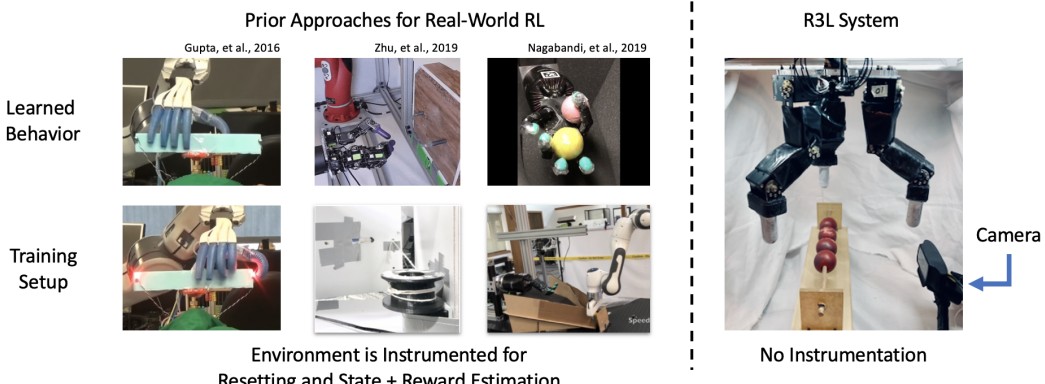

Figure 2: We draw a comparison between current real world learning systems which rely on instrumentation versus a system that learns in an environment more representative of the real world, free of instrumentation. While all three prior works utilize instrumentation for resets, state estimation, and reward, the motion capture system of Gupta et al. (2016), sensor attached to the door in Zhu et al. (2019), and auxiliary robot which picks up fallen balls in Nagabandi et al. (2019) are good examples of engineered state estimation, reward estimation, and reset mechanisms respectively.

## 2.1 Learning from Raw Sensory Input

To enable learning without complex state estimation systems or environment instrumentation, we require our robotic systems to be able to learn from their own raw sensory observations. Typically, these sensory observations are raw camera images from a camera mounted on the robot, as well as proprioceptive sensory inputs such as the joint angles. These observations do not directly provide the poses of the objects in the scene, which is the typical assumption in simulated robotic environments – any such information must be extracted by the learning system.

While in principle many RL frameworks can support learning from raw sensory inputs (Levine et al., 2016; Mnih et al., 2015; Schulman et al., 2015; Lillicrap et al., 2015), it is important to consider the practicalities of this approach. For instance, we can instantiate vision-based RL with policy gradient algorithms such as TRPO (Schulman et al., 2015) and PPO (Schulman et al., 2017), but these have high sample complexities which make them unsuited for real world robotic learning (Haarnoja et al., 2018a). In our work, we consider adopting the general framework of off-policy actor-critic reinforcement learning, using a version of the soft actor critic (SAC) algorithm described by (Haarnoja et al., 2018b). This algorithm effectively uses off-policy data and has been shown to learn some tasks directly from visual inputs. However, while SAC can learn directly from images, we find in our experiments that, as the task complexity increases, the efficiency of direct end-to-end learning (particularly without resets and with learned rewards) still degrades substantially. However, as we will discuss in Section 3, augmenting end-to-end learning with unsupervised representation learning substantially alleviates such challenges.

## 2.2 Reward Functions without Reward Engineering

Vision-based RL algorithms, such as SAC, rely on a reward function being provided to the system, which is typically manually programmed by a user. While this can be simple to do in simulation by using ground truth state information, it is significantly harder to implement in uninstrumented real world environments. In the real world, the robot must obtain the reward signal itself from its own sensor readings. A few options for tackling this challenge have been discussed in prior work: design complete computer vision systems to detect objects and extract the reward signals (Devin et al., 2018; Nagabandi et al., 2019), engineer reward functions that use various task-specific heuristics to obtain rewards from pixels (Schenck & Fox, 2017; Kalashnikov et al., 2018), or instrument every environment (Chebotar et al., 2017). Many of these solutions are manual and tedious, and a more general approach is needed to scale real-world robotic learning gracefully.

## 2.3 LEARNING WITHOUT RESETS

While the components described in Section 2.1 and 2.2 are essential to building continuously learning RL systems in the real world, they have often been implemented with the assumption of episodic learning Fu et al. (2018); Haarnoja et al. (2018b). However, natural open-world settings do not provide any such reset mechanism, and in order to enable scalable and autonomous real-world learning we need systems that do not require an episodic formulation of the learning problem.

To devise a system that requires minimal human engineering for providing rewards, we must use algorithms that are able to assign *themselves* rewards, using learned models that operate on the same raw sensory inputs as the policy. One candidate is for a user to specify intended behavior beforehand through examples of desired outcomes (i.e., images). The algorithm can then assign itself rewards based on a measure of how well it is accomplishing the specified goals, with no additional human supervision. This approach can scale well in principle, since it requires minimal human engineering, and goal images are easy to provide.

In principle, algorithms such as SAC do not actually require episodic learning; however, in practice, most instantiations use explicit resets, even in simulation, and removing resets has resulted in failure to solve challenging tasks. In our experiments in Section 3 as well, we see that actor-critic methods applied naïvely to the reset free setting do not learn the intended behaviors. Introducing visual observations and classifier based rewards exacerbates these challenges.

We propose that these three components – vision-based RL with actor-critic algorithms, vision-based goal classifier for rewards, and reset-free learning – are the fundamental pieces that we need to build a real world robotic learning system. However, when we actually combine the individual components in Sections 3 and 6, we find that learning effective policies is quite challenging. We provide insight into these challenges in Section 3. Based on these insights, we propose simple but important changes in Section 4 to build a system, R3L , that can learn effectively and autonomously in the real world without human intervention.

## 3 THE CHALLENGES OF REAL WORLD RL

The system design outlined in Section 2 in principle gives us a complete system to perform real world reinforcement learning without instrumentation. However, when utilized for robotic learning problems, we find this basic design to be largely ineffective. To study this, we present results for a simulated robotic manipulation task that requires repositioning a free-floating objects with a three-fingered robotic hand, shown in Fig 3. We use this task for our investigative analysis and show that the same insights extend to several other tasks, including real world tasks, in Section 6. The goal in this task is to reposition the object to a target pose from any initial pose in the arena. When the system is instantiated with vision-based soft actor-critic, rewards from goal images using VICE, and run without episodic resets, we see that the algorithm fails to make progress (Fig 4). Although it might appear that this setup fits within the assumptions of all of the components that are used, the complete system is ineffective. Which particular components of this problem make it so difficult?

Figure 3: Our object repositioning task. The goal is to move the object from any starting configuration to a particular goal position and orientation.

To investigate this issue, we perform experiments investigating the combination of the three main ingredients: varying observation type (visual vs. low-dimensional state), reward structure (VICE vs. hand-defined rewards that utilize ground-truth object state), and the ability to reset (episodic resets vs. reset-free, non-episodic learning). We start by considering the training time reward under each combination of factors as shown in Fig 4, which reveals several trends. First, the results in Fig 4 show that learning with resets achieves high training time reward from both vision and state, while reset-free only achieves high training time reward with low-dimensional state. Second, we find that the policy is able to pass the threshold for training time reward in a non-episodic setting when learning from low-dimensional state, but it

| VICE True Reward | With Resets | Without Resets |
|---|---|---|
| State | 700k / 200k | 1M / 500k |
| Vision | ✗ / 800k | ✗ / ✗ |

Figure 4: We report the approximate number of samples needed for a policy learned with a prior *off-policy RL algorithm (SAC)* to achieve average training performance of less than 0.15 in pose distance (defined in Appendix C.1.3) across 3 seeds on the re-positioning task. We compare training performance after varying three axes: ground truth rewards vs. learned rewards, with vs. without episodic resets, low-level state vs. images as inputs. We observe learning without resets is harder than with resets and is much harder when combined with visual inputs.

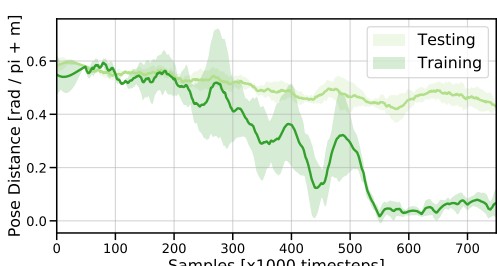

Figure 5: We observe that when training reset free to reach a single goal, while the pose distance at training time is quite low, the pose errors obtained at test-time with the learned policy are very high. This indicates that while the object is getting close to the goal at training time, the policies being learned are still not effective.

is not able to do the same using image observations. This suggests that combining the reset-free learning problem with visual observations makes it significantly more challenging.

However, the table in Fig 4 paints an incomplete picture. These numbers are related to the performance of the policies at training time, not how effective the learned policies are when being evaluated. When we consider the test-time performance (Fig 5) of the policies that are learned under reset free conditions, we obtain a different set of conclusions. While learning from low-dimensional state in the reset free setting achieves high reward at training time, the test-time performance of the corresponding learned policies is very poor. This can likely be attributed to the fact that when the agent spends all its time stuck at the goal, and sees very little diversity of data in other parts of the state space, which significantly reduces the efficacy of the actual policies being learned. In a sense, the reset encodes some prior information about the task: it tells the policy about what types of states it might be required to succeed from at test time. Without this knowledge, performance is substantially worse. This makes it very challenging to learn policies with completely reset-free schemes, which has prompted prior work to consider schemes such as learning reset controllers (Eysenbach et al., 2018). As we discuss in the following section and in our experiments, these schemes are often insufficient for learning effective policies in the real world without *any* resets.

# 4 A REAL-WORLD ROBOTIC REINFORCEMENT LEARNING SYSTEM

To address the challenges identified in Section 3, we present two improvements, which we found to be essential for uninstrumented real-world training: randomized perturbation controllers and unsupervised representation learning. Incorporating these components into the system in Section 2 results in a system that can learn successfully in uninstrumented environments, as we will show in Section 6, and attains good performance both at training time and at test time.

## 4.1 RANDOMIZED PERTURBATION CONTROLLER

Prior works in addressing the reset free setting problem have considered converting the problem into a more standard episodic problem, by learning a "reset controller," which is trained to reset the system to a particular *initial state* (Eysenbach et al., 2018; Han et al., 2015). This scheme has been thought to make the learning problem easier by reducing the variance of the resulting initial state distribution. However, as we will show in our experiments in Section 6, this still results in policies whose success depends heavily on a narrow range of initial states. Indeed, prior reset controller methods all reset to a *single* initial state (Eysenbach et al., 2018; Han et al., 2015).

We take a different approach to learning in a reset-free setting. Rather than attributing the problem to the variance of the initial state distribution, we hypothesize that a major problem with reset-free learning is that the support of the distribution of states visited by the policy is too narrow, which

makes the learning problem challenging and doesn't allow the agent to learn how to perform the desired task from *any* state it might find itself in. In this view, the goal should not be to *reduce* the variance of the initial state distribution, but instead to *increase* it.

To this end, we utilize what we call random perturbation controllers: controllers that introduce perturbations intermittently into the system through a policy that is trained to explore the environment. The standard actor $\pi(a|s)$ is executed for $H$ time-steps, following which we executed the perturbation controller $\pi_p(a|s)$ for $H$ steps, and repeat. The policy $\pi$ is trained with the VICE-based rewards for reaching the desired goals, while the perturbation controller $\pi_p$ is trained only with an intrinsic motivation objective that encourages visiting under-explored states. In our implementation, we use the random network distillation (RND) objective for training the perturbation controller (Burda et al., 2018), but any effective exploration method can be used for the same. This procedure is described in detail in Appendix A, and is evaluated on the tasks we consider in Fig 6. The perturbation controller ensures that the support of the training distribution grows and as a result the policies can learn the desired behavior much more effectively, as shown in Fig 7.

## 4.2 GOAL CLASSIFIER

To design a system that requires minimal human engineering for providing reward, we use a data-driven reward specification framework called variational inverse control with events (VICE) introduced by Fu et al. (2018). VICE learns rewards in a task-agnostic way: we provide the algorithm with success examples in the form of images where the task is accomplished, and learn a discriminator that is capable of distinguishing successes from failures. This discriminator can then be used to provide a learning signal to nudge the reinforcement learning agent towards success. This algorithm has been previously considered in the context of learning tasks from raw sensory observations in the real world by (Singh et al., 2019) but we show that it presents unique challenges when used in conjunction with learning without episodic resets. Details and specifics of the algorithms being considered are described in Appendix A and also discussed by Singh et al. (2019).

## 4.3 UNSUPERVISED REPRESENTATION LEARNING

The perturbation controller discussed above allows us to learn policies that can succeed at the task from a variety of starting states. However, learning from visual observations still present a challenge. Our experiments in Fig 4 show that learning without resets from low-dimensional states is comparatively easier. We therefore aim to convert the vision-based learning problem into one that more closely resembles state-based learning, by training a variational autoencoder (VAE, Kingma & Welling (2013)) and sharing the latent-variable representation across the actor and critic networks (refer to Appendix B for more details). Note that we use a VAE as an instantiation of representation learning techniques that works well in the domains we considered, but other more sophisticated density models proposed in prior work may also be substituted in place of the VAE (Lee et al., 2019; Hjelm et al., 2019; Anand et al., 2019).

While several works have also sought to incorporate unsupervised learning into reinforcement learning to make learning from images easier (Nair et al., 2018; Lee et al., 2019), we note that this becomes especially critical in the vision-based, reset-free setting, as motivated by the experiments in Section 3, which indicate that it is precisely this combination of factors – vision and no resets – that presents the most difficult learning problem. Therefore, although the particular solution we use in our system has been studied in prior work, it is brought to bare to address a challenge that arises in real-world learning that we believe has not been explored in prior studies.

These two improvements – the perturbation controller and unsupervised learning – combined with the general system described above, give us a complete practical system for real world reinforcement learning. The overall method uses soft-actor critic for learning with visual observations and classifier based rewards with VICE, introduces auxiliary reconstruction objectives or pretrains encoders for unsupervised representation learning, and uses a perturbation controller during training to ensure that the support of visited states grows sufficiently. We term this full system for real-world robotic reinforcement learning R3L. Further implementation details can be found in Appendix A.

## 5 RELATED WORK

The primary contribution of this work is to propose a paradigm for continual instrumentation-free real world robotic learning and a practical instantiation of such a system. A number of prior works have studied reinforcement learning in the real world for acquiring robotic skills (Levine et al., 2016; Kumar et al., 2016; Gu et al., 2017; Kalashnikov et al., 2018; Haarnoja et al., 2018b; Finn & Levine, 2016; Zhu et al., 2019; Nagabandi et al., 2019). Much of the focus in prior work has been on improving the efficiency and generalization of RL algorithms to make real-world training feasible, or else on utilizing simulation and transferring policies into the real world (Tzeng et al., 2015; Tobin et al., 2017; Peng et al., 2017; Clavera et al., 2018; Kang et al., 2019). The simulation-based methods typically require considerable effort in terms of both simulator design and overcoming the distributional shift between simulated and real-experience, while prior real-world training methods typically require additional instrumentation for either reward function evaluation (Levine & Koltun, 2013) or resetting between trials (Gu et al., 2017; Chebotar et al., 2017; Zhu et al., 2019), or both. In contrast, our work is primarily focused on lifting these requirements, rather than devising more efficient RL methods. We show that removing the need for instrumentation (i.e., for reward evaluation and resets) introduces additional challenges, which in turn require a careful set of design choices. Our complete R3L method is able to learn completely autonomously, without manual resets or reward design.

A key component of our system involves learning from raw visual inputs. This has proven to be difficult for policy gradient style algorithms (Pinto et al., 2017a) due to challenging representation learning problems. This has been made easier in simulated domains by using modified objectives, such as auxiliary losses (Jaderberg et al., 2016), or by using more efficient algorithms (Haarnoja et al., 2018a). We show that reinforcement learning on raw visual input, while possible in standard RL settings, becomes significantly more challenging when considered in conjunction with non-episodic, reset-free scenarios.

Reward function design is crucial for any RL system, and is non-trivial to provide in the real world. Prior works have considered instrumenting the environment with additional sensors to evaluate rewards (Gu et al., 2017; Chebotar et al., 2017; Zhu et al., 2019), which is a highly manual process, using demonstrations, which require manual effort to collect (Vecerik et al., 2017; Ng & Russell, 2000; Liu et al., 2018), or using interactive supervision from the user (Christiano et al., 2017). In this work, we leverage the algorithm introduced by Fu et al. (2018) to assign rewards based on the likelihood of a goal classifier. While prior work also applied this method to robotic tasks (Singh et al., 2019), this was done in a setting where manual resets were provided by hand, while we demonstrate that we can use learned rewards in a fully uninstrumented, reset-free setup.

Learning without resets has been considered in prior works (Eysenbach et al., 2018; Han et al., 2015), although in different contexts – safe learning and learning compound controllers, respectively. Eysenbach et al. (2018) provide an algorithm to learn a reset controller with the goal of ensuring safe operation, but makes several assumptions that make it difficult to use in the real world: it assumes access to a ground truth reward function, it assumes access to an oracle function that can detect if an attempted reset by the reset policy was successful or not, and it assumes the ability to perform manual resets if the reset policy fails a certain number of times. In contrast, we propose an algorithm that allows for fully automated reinforcement learning in the real world. We compare to an ablation of our method that uses a reset controller similar to Eysenbach et al. (2018), and show that our method performs substantially better. Our perturbation controller also resembles the adversarial RL setup Pinto et al. (2017b); Sukhbaatar et al. (2018). However, while these prior methods explicitly aim to train policies that are robust to perturbations Pinto et al. (2017b) or explore effectively Sukhbaatar et al. (2018), we are concerned with learning without access to resets.

While this line of work has connections to developmental robotics (Lungarella et al., 2003; Asada et al., 2009) and its subfields, such as continual (Lesort et al., 2019) and lifelong (Thrun & Mitchell, 1995) learning, the goal of our work is to handle the practicalities of enabling reinforcement learning systems to learn in the real world without instrumentation or interruption, even for a single task setting. Though our work does not directly study continual lifelong learning, nor all facets of developmental robotics, it relates to continual learning (Lesort et al., 2019), intrinsic motivation (Schmidhuber, 2006) and sensory-motor development involving proprioceptive manipulation (Stoica, 2001).

# 6 EXPERIMENTS

In our experimental evaluation, we study how well the R3L system, described in Sections 2 and 4, can learn under realistic settings – visual observations, no hand-specified rewards, and no resets. We consider the following hypotheses:

1. Can we use R3L to learn complex robotic manipulation tasks without instrumentation? Does this system learn skills in both simulation and the real world?

2. Do the solutions proposed in Section 4 actually enable R3L to perform tasks without instrumentation that would not have been otherwise possible?

## 6.1 EXPERIMENTAL SETUP

We consider the task of dexterous manipulation with a three fingered robotic hand, the D'Claw (Zhu et al., 2019; Ahn et al., 2019), on a number of simulated and real world tasks. These tasks involve complex coordination of three fingers with 3 DoF each in order to manipulate objects. Prior works that used this robot utilized explicit resets and low-dimensional true state observations, while we consider settings with visual observations, no hand-specified rewards, and no resets.

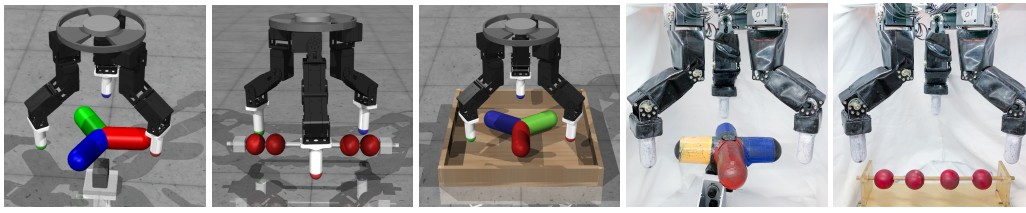

Figure 6: Visualizations of the goal configurations of the simulated and real world tasks being considered. From left to right we depict valve rotation, bead manipulation and free object repositioning in simulation, as well as valve rotation and bead manipulation manipulation in the real world.

The tasks in our experiments are shown in Fig 6: manipulating beads on an abacus row, valve rotation, and free object repositioning. These tasks represent a wide class of problems that robots might encounter in the real world. For each task, we consider the problem of reaching the depicted goal configuration: moving the abacus beads to a particular position, rotating the valve to a particular angle, and repositioning the free object to a particular goal position. For each task, policies are evaluated from a wide selection of initial configurations. Additional details about the tasks and evaluation procedures are provided in Appendix C. Videos and additional details can be found at https://sites.google.com/view/realworld-rl/

## 6.2 LEARNING IN SIMULATION WITHOUT INSTRUMENTATION

We compare our entire proposed system implementation (Section 4) with a number of baselines and ablations. Importantly, all methods must operate under the same assumptions: none of the algorithms have access to system instrumentation for state estimation, reward specification, or episodic resets. Firstly, we compare the performance of R3L to a system which uses SAC for vision-based RL from raw pixels, VICE for providing rewards and running reset-free (denoted as "VICE"). This corresponds to the vanilla version of R3L (Section 2), with none of the proposed insights and changes. We then compare with prior reset-free RL algorithms (Eysenbach et al., 2018) that explicitly learn a reset controller to alternate goals in the state space ("Reset Controller + VAE"). Lastly, we compare algorithm performance with two ablations: running R3L without the perturbation controller ("VICE + VAE") and without the unsupervised learning ("R3L w/o VAE"). This highlights the significance of each of the components of R3L .

From the experimental results in Fig 7, it is clear that R3L is able to reach the best performance across tasks, while none of the other methods are able to solve all of the tasks. Different prior methods and ablations fail for different reasons: (1) methods without the reconstruction objective struggle at parsing the high-dimensional input and are unable to solve the harder task; (2) methods

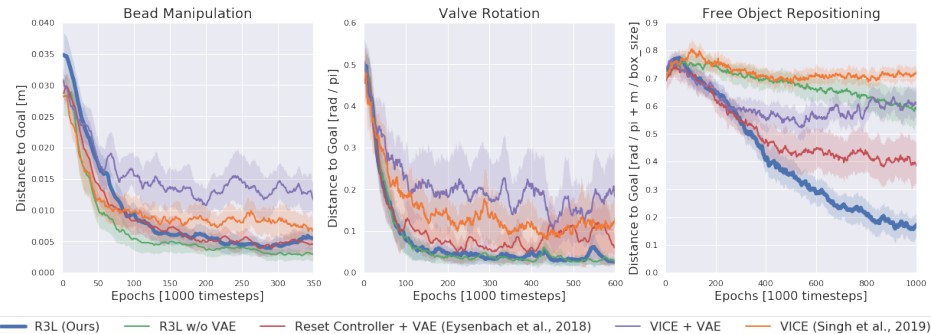

Figure 7: Quantitative evaluation of performance in simulation for bead manipulation, valve rotation and free object repositioning (left to right) run with 10 random seeds. The error bars show 95% bootstrap confidence intervals for average performance. While other variants are sufficient to get good evaluation performance on easier tasks, harder tasks like free object repositioning require random perturbations and unsupervised representation learning to learn skills reset-free. See Appendix C.1 for details of evaluation procedures.

without the perturbation controller are ineffective at learning how to reach the goal from novel initialization positions for the more challenging object repositioning tasks, as discussed in Section 4.

We note that an explicit reset controller, which can be viewed as a softer version of our perturbation controller with goal-directedness, learns to solve the easier tasks due to the reset controller encouraging exploration of the state space. In our experiments for free object repositioning, performance was reported across 3 choices of reset states. The high variance in evaluation performance indicates that the performance of such a controller (or a goal conditioned variant of it) is highly sensitive to the choice of reset states. A poor choice of reset states, such as two that are very close together, may yield poor exploration leading to performance similar to the single goal VICE baseline. Furthermore, the choice of reset states is highly task dependent and it is often not clear what choice of goals will yield the best performance. On the contrary, our method does not require such task-specific knowledge and uses random perturbations to reset while training without any explicit reset states: this allows for a robust, instrumentation-free controller while also ensuring fast convergence.

### 6.3 Learning in the Real World without Instrumentation

Since the aim of R3L is to enable uninstrumented training in the real world, we next evaluate our method on a real-world robotic system, providing evidence that our insights generalize to the real world *without any instrumentation*. After providing the initial outcome examples for learning the reward function with VICE, we leave the robot unattended, and the algorithm learns the desired behavior through interaction. The experiments are performed on the D'Claw robotic hand with an RGB camera as the only sensory input. Intermediate policies are saved at regular intervals, and evaluations of all policies is performed after training has completed. For valve rotation, we declare an evaluation rollout a success if the final orientation is within within 15° of the goal. For bead manipulation, we declare success if all the beads are within 2cm of the goal state. Fig 8 compares the performance of our method without supervised learning ("R3L w/o VAE") in the real world against a baseline that uses SAC for vision-based RL from raw pixels, VICE for providing rewards, and running reset-free (denoted as "VICE"). We see that our method learns policies that succeed from nearly all the initial configurations, whereas VICE alone fails from most initial configurations. Fig 9 depicts sample evaluation rollouts of the policies learned using our method. For further details about real world experiments see Appendix C.2.

## 7 Discussion

We presented the design and instantiation of a system for real world reinforcement learning. We identify and investigate the various ingredients required for such a system to scale gracefully with minimal human engineering and supervision. We show that this system must be able to learn from raw sensory observations, learn from very easily specified reward functions without reward engineering, and learn without any episodic resets. We describe the basic elements that are required

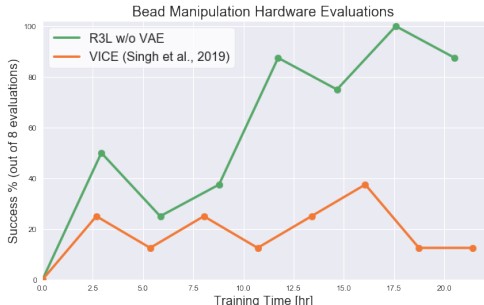
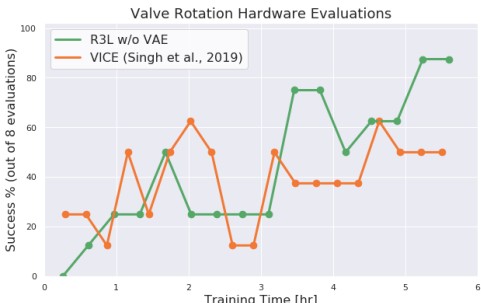

Figure 8: Quantitative evaluation of performance in real-world for valve rotation and bead manipulation. Policies trained with perturbation controllers have effectively learned behaviors after 17 and 5 hours of training, respectively. For more fine-grained reporting of results see Figs 13-16.

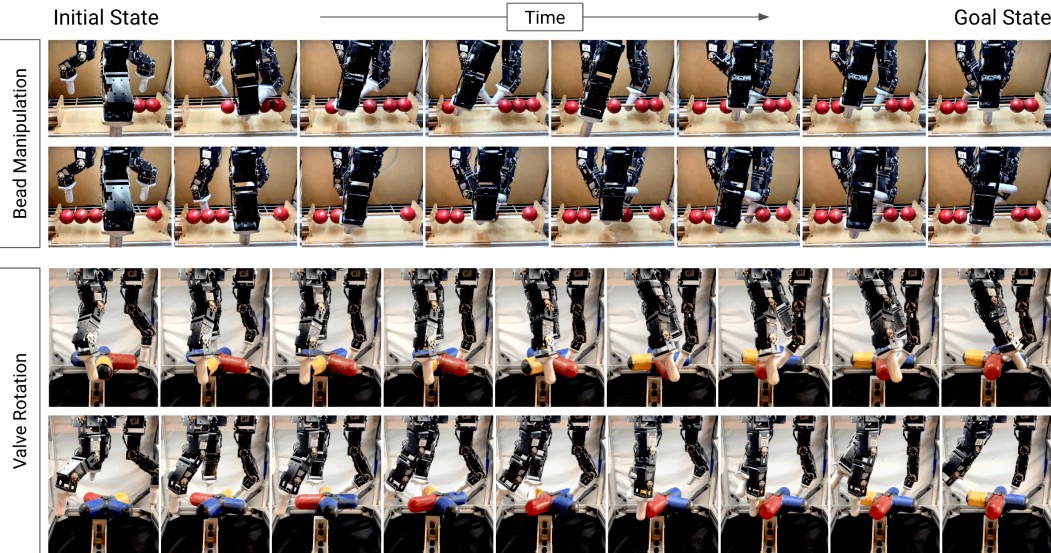

Figure 9: Evaluation rollouts of R3L on the real world tasks for policies trained without instrumentation. Successful evaluation rollouts for bead manipulation (top) and valve rotation (bottom) are shown here.

to construct such a system, and identify unexpected learning challenges that arise from interplay of these elements. We propose simple and scalable fixes to these challenges through introducing unsupervised representation learning and a randomized perturbation controller.

The ability to train robots directly in the real world with minimal instrumentation opens a number of exciting avenues for future research. Robots that can learn unattended, without resets or hand-designed reward functions, can in principle collect very large amounts of experience autonomously, which may enable very broad generalization in the future. However, there are also a number of additional challenges, including sample complexity, optimization and exploration difficulties on more complex tasks, safe operation, communication latency, sensing and actuation noise, and so forth, all of which would need to be addressed in future work in order to enable truly scalable real-world robotic learning.

## ACKNOWLEDGMENTS

This research is supported by the Office of Naval Research, the National Science Foundation under IIS-1651843 and IIS-1700696, Google, NVIDIA, and Amazon. The authors would like to thank Ignasi Clavera, Gregory Kahn, Coline Devin, Benjamin Eysenbach, Aviral Kumar, Anusha Nagabandi, Marvin Zhang, Ashvin Nair and several others in the RAIL Lab for helpful discussions and feedback.

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

## A ALGORITHM DETAILS

---

**Algorithm 1** Real-World Robotic Reinforcement Learning (R3L)

---

1: **procedure** R3L
2:     $N \leftarrow$ number of training epochs
3:     $H \leftarrow$ trajectory length (horizon)
4:     $n_{\text{VICE}} \leftarrow$ number of VICE classifier training iterations per epoch
5:     Initialize forward and perturbing policies $\pi_0, \pi_1$
6:     Obtain goal states $s_i^E$ and initialize as a goal pool $\mathcal{G}$
7:     Initialize RND target and predictor networks $f(s), \hat{f}(s)$
8:     Initialize VICE reward classifier $r_{\text{VICE}}(s)$
9:     Initialize replay buffer $\mathcal{D}$
10:     Collect initial exploration data and add to $\mathcal{D}$
11:     **for** $i = 1$ to $2N$ **do**
12:         $k \leftarrow i \% 2$
13:         **for** $t = 1$ to $H$ **do**
14:             Sample $a_t \sim \pi_k(s_t)$
15:             Sample $s_{t+1} \sim p(s_{t+1}|s_t, a_t)$
16:             **if** k == 0 **then**
17:                 $r_t(s_t) = c_{\text{VICE}} * r_{\text{VICE}}(s_t) + c_{\text{RND}} * r_{\text{RND}}(s_t)$
18:             **else if** k == 1 **then**
19:                 $r_t(s_t) = r_{\text{RND}}(s_t)$
20:             **end if**
21:             Sample batch from $\mathcal{D}$
22:             Update $\pi_k$ with batch according to SAC
23:             Update RND predictor network with batch
24:             Update running estimate of standard deviations of classifier and RND reward
25:         **end for**
26:         Add experience to the replay buffer with $\mathcal{D} \leftarrow \mathcal{D} \cup \tau_i$
27:         Sample an equal number of goal examples from $\mathcal{G}$ and negative examples from $\mathcal{D}$
28:         **for** $t = 1$ to $n_{\text{VICE}}$ **do**
29:             Train the VICE classifier on this batch with binary labels
30:         **end for**
31:     **end for**
32: **end procedure**

---

# B  Training details

## B.0.1  Hyperparameters

| General | |
|---|---|
| Standard deviation update coefficient | **0.99** |
| Image Sizes | [(16, 16, 3), **(32, 32, 3)**, (64, 64, 3)] |
| **SAC** | |
| Learning Rate | **3e-4** |
| $\gamma$ | **0.99** |
| Batch Size | **256** |
| Convnet Filters | [**(64, 64, 64)**, (16, 32, 64)] |
| Stride | **(2, 2)** |
| Kernel Sizes | **(3, 3)** |
| Pooling | [MaxPool2D, **None**] |
| Actor/Critic FC Layers | [**(512, 512)**, (256, 256, 256)] |
| **VICE** | |
| $n_{\text{VICE}}$ | [1, **5**, 10] |
| Batch Size | **128** |
| Learning Rate | **1e-4** |
| Mixup $\alpha$ | **Uniform(0, 1)** |
| Convnet Filters | [**(64, 64, 64)**, (16, 32, 64)] |
| Stride | **(2, 2)** |
| Kernel Sizes | **(3, 3)** |
| Pooling | [MaxPool2D, **None**] |
| FC Layers | [**(512, 512)**, (256, 256, 256)] |
| **RND** | |
| Learning Rate | **3e-4** |
| Batch Size | **256** |
| Convnet Filters | **(16, 32, 64)** |
| Stride | **(2, 2)** |
| Kernel Sizes | **(3, 3)** |
| Pooling | [MaxPool2D, **None**] |
| FC Layers | [**(512, 512)**, (256, 256, 256)] |
| **VAE** | |
| Learning Rate | **1e-4** |
| Batch Size | **256** |
| Encoder (Convnet) Filters | **(64, 64, 32)** |
| Latent Dimension | [8, **16, 32**, 64] |
| $\beta$ | [1e-3, 0.1, **0.5**, 1, 10] |
| Stride | **(2, 2)** |
| Kernel Sizes | **(3, 3)** |
| Pooling | [MaxPool2D, **None**] |

The ranges of values listed above represent the hyperparameters we searched over, and the bolded values are what we use in the Section 6 experiments.

## B.0.2  VICE

We use a variant of VICE which defines the reward as the logits of the classifier, notably omitting the $-log(\pi(a|s))$ term. We also regularize our classifier with mixup (Zhang et al., 2018). We train all of our experiments using 200 goal images, which takes under an hour to collect in the real world for each task.

## B.0.3  Random Network Distillation (RND)

We found it important to normalize the predictor errors, just as (Burda et al., 2018) did.

### B.0.4 VAE

We train a standard beta-VAE to maximize the evidence lower bound, given by:

$$\mathbb{E}_{z \sim q_\phi(z|x)}[p_\theta(x|z)] - \beta D_{\text{KL}}(q_\phi(z|x) \,||\, p_\theta(z))$$

To collect training data, we sampled random states in the observation space. In the real world, this sampling can be replaced with training an exploratory policy (i.e. using the RND reward as the policy's only objective). The learned weights of the encoder of the VAE are frozen, and the latent input is used to train the policy for reset-free RL.

## C TASK DETAILS

### C.1 SIMULATED TASKS

We evaluated our system across three tasks in simulation: bead manipulation, valve rotation, free object repositioning.

#### C.1.1 BEAD MANIPULATION

The bead manipulation task involves an abacus rod with four beads that can slide freely. The goal is to position two beads on each end from any initial configuration of beads. This can take the form of sliding one bead over (if three beads start on one side), two beads over (if all four beads start on one side), splitting beads apart (all four beads start in the middle), or some intermediate combination of those. The true reward is defined as the mean goal distance of all four beads. Policies are evaluated starting from the 8 initial configurations depicted in Fig 10. Evaluation performance reported in Section 6 for this task is defined as the final reward averaged across the 8 evaluation rollouts.

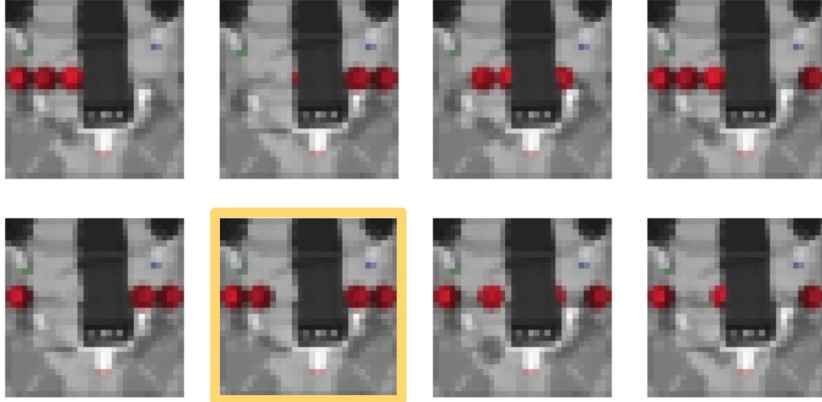

Figure 10: These are the 8 initial positions used for evaluating the performance of the bead manipulation policy. The goal configuration (which is also an initial evaluation position) is highlighted in yellow.

#### C.1.2 VALVE ROTATION

The claw is positioned above a three pronged valve (15 cm in diameter). The objective is to turn the valve to a given orientation from any initial orientation. The "true reward" is $r = -\log(|\theta_{state} - \theta_{goal}|)$. Policies are evaluated starting from the 8 initial configurations depicted in Fig 11. Evaluation performance reported in Section 6 for this task is defined as the final orientation distance averaged across the 8 evaluation rollouts.

#### C.1.3 FREE OBJECT REPOSITIONING

The claw is positioned atop a free (6 DoF) three pronged object (15cm in diameter), which can translate and rotate within a 30cm$x$30cm box. The goal is specified by a xy-position as well as a z-angle, where the xy-plane is the plane of the arena. The true reward is defined as the weighted sum of the angular and translational distances, $r = -2\log(||[x_{state}, y_{state}] - [x_{goal}, y_{goal}]||_2) -$

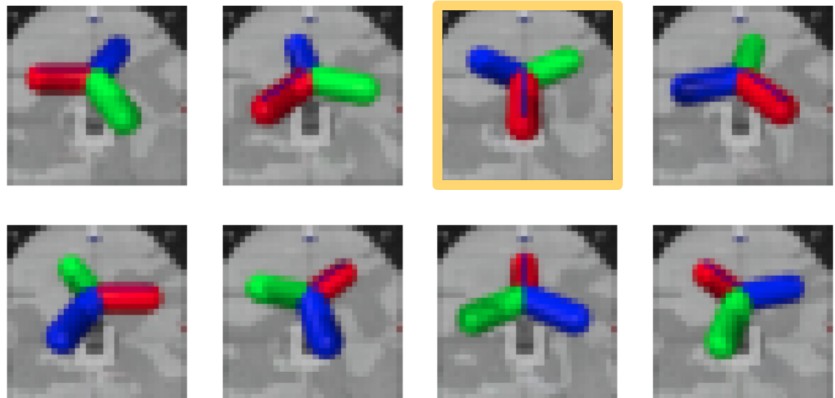

Figure 11: These are the 8 initial positions used for evaluating the performance of the valve rotation policy. The goal configuration (which is also an initial evaluation position) is highlighted in yellow.

$\log(|\theta_{state} - \theta_{goal}|)$. In our experiments, $(x, y, \theta)_{goal} = (0, 0, -\frac{\pi}{2})$, where the origin is centered in the arena. Policies are evaluated starting from the 15 initial configurations depicted in Figure 12. Evaluation performance reported in Section 6 for this task is defined as the final pose distance ($\frac{||[x_{final}, y_{final}] - [x_{goal}, y_{goal}]||_2}{0.25 \text{ m}} + \frac{|\theta_{final} - \theta_{goal}|}{\pi \text{ rad}}$) averaged across the 15 evaluation rollouts.

In our reset controller experiments, we averaged evaluation performance over three different choices of reset states, where the first reset state is always the goal:

1. $(x, y, \theta)_{reset,1} = (x, y, \theta)_{goal}, (x, y, \theta)_{reset,2} = (0.05, 0.05, \frac{\pi}{2})$

2. $(x, y, \theta)_{reset,1} = (x, y, \theta)_{goal}, (x, y, \theta)_{reset,2} = (0, 0, -\frac{\pi}{6})$

3. $(x, y, \theta)_{reset,1} = (x, y, \theta)_{goal}, (x, y, \theta)_{reset,2} = (-0.04, -0.04, -\frac{\pi}{2})$

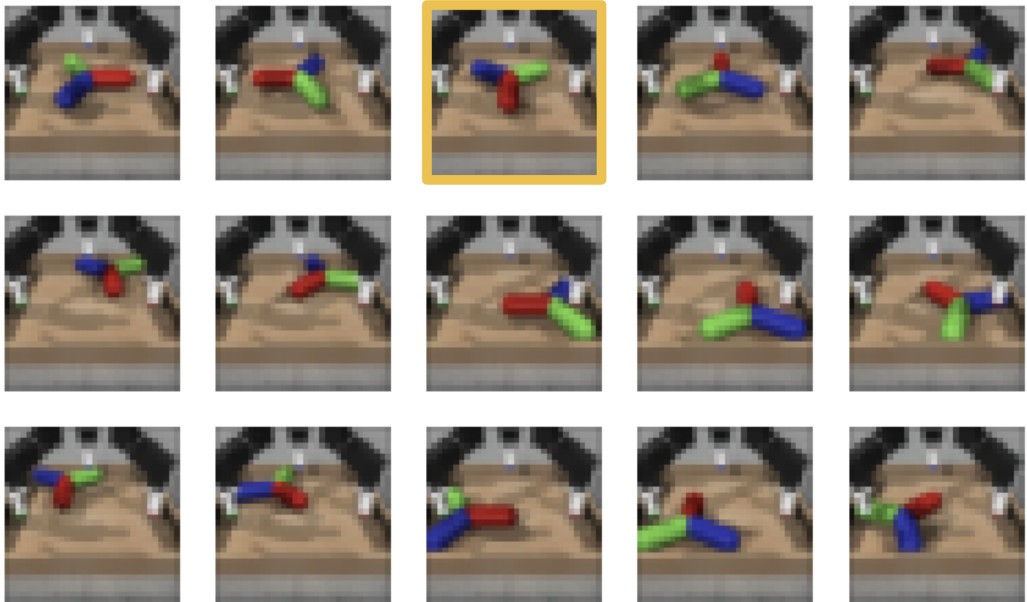

Figure 12: These are the 15 initial positions used for evaluating the performance of the free object repositioning policy. The goal configuration $(x, y, \theta)_{goal}$ which is also an initial evaluation position is highlighted in yellow.

## C.2 REAL WORLD TASKS

For each setup we use an RGB camera to get images. We execute actions on the DClaw at 10Hz. In order to operate at such a high frequency while also training from images we sample and train asynchronously, but limit training to not exceed two gradient steps per transition sampled in the real world. Since direct performance metrics cannot be measured during training due to the lack of object instrumentation, evaluations of performance are done post-training.

### C.2.1 VALVE ROTATION

The task is identical to the one in simulation. Evaluations were done post-training. An evaluation trajectory was defined as a success if at the last step, the valve was within 15 degrees of the goal. Each policy was evaluated over 8 rollouts, with initial configurations evenly spaced out between at increments of 45 degrees. Results are reported in Figures 13, 14.

### C.2.2 BEAD MANIPULATION

The rod is 22cm in length, and each bead measures 3.5cm in diameter. Evaluations were done post-training, using the following procedure: the environment was manually reset to each of the 8 specified configurations shown in Figures 15 and 16 (which cover a full range of the state space) at the start of each evaluation rollout. An evaluation trajectory was defined as a success if at the last time step, all beads were within 2cm of their goal positions. Performance was evaluated at around 20 hours, at which point the policy achieved greater than 80% success on the 10 evaluation rollouts (a random policy achieved a success rate of 10%). Results are reported in Figs 15, 16.

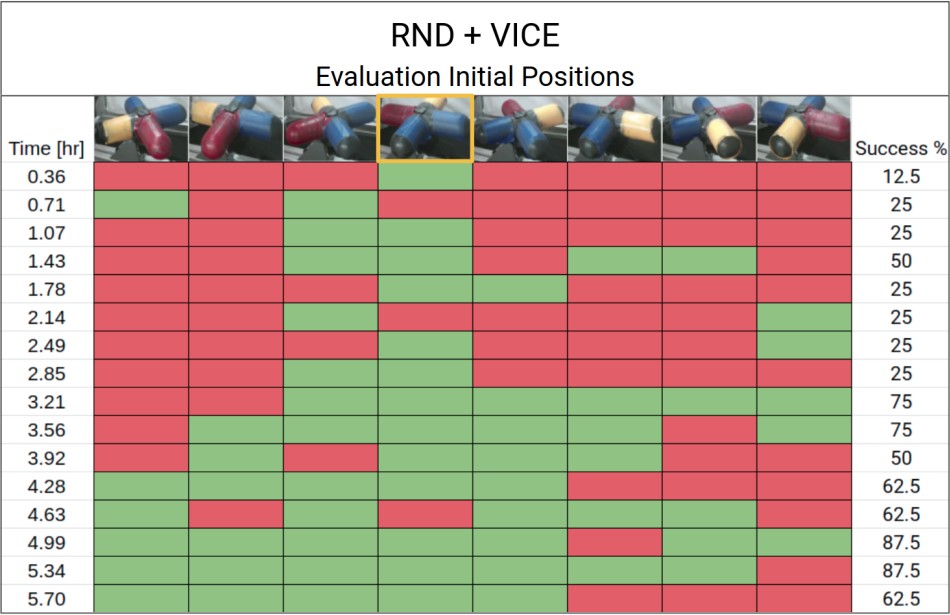

Figure 13: These are the results of the evaluation rollouts on the valve rotation task in the real world using our method (without the VAE). Trained policies were saved at regular intervals and evaluated post-training. Each row is a different policy, and each column an evaluation rollout from a different initial configuration. The goal is highlighted in yellow. Our method is able to achieve high success rates after 5 hours of training.

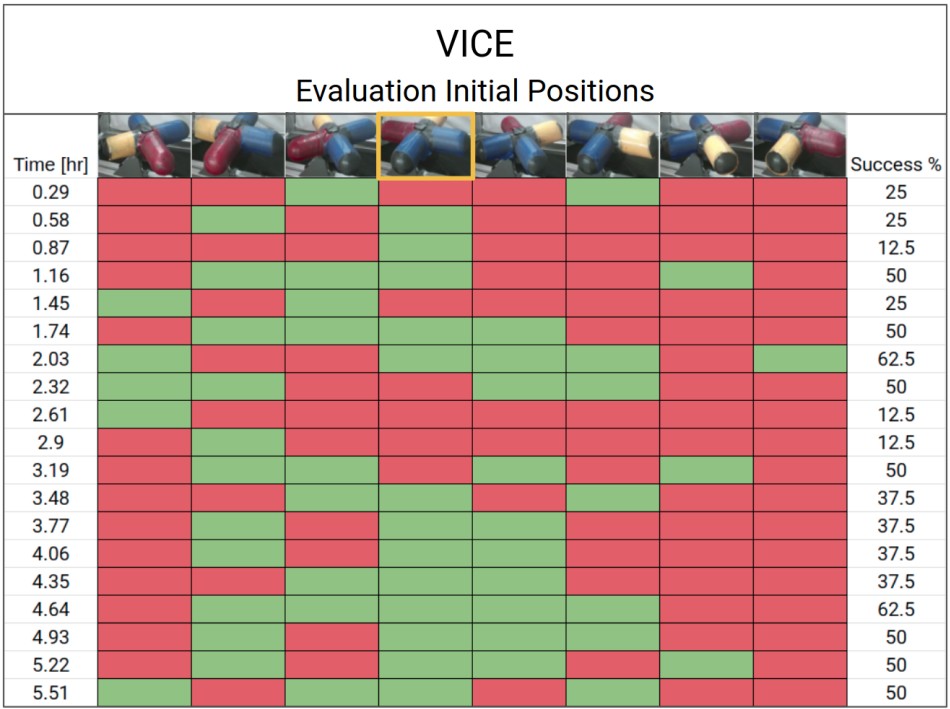

Figure 14: These are the results of evaluation rollouts on the valve rotation task in the real world using the VICE single goal baseline. The policies fail to evaluate well, especially from initial positions far from the goal position.

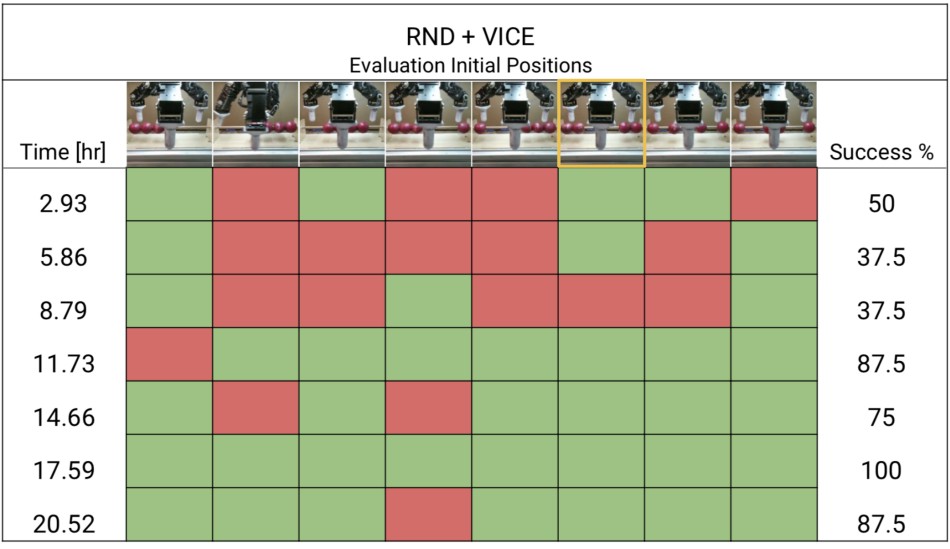

Figure 15: These are the results of the evaluation rollouts on the valve rotation task in the real world using our method (without the VAE). Trained policies were saved at regular intervals and evaluated post-training. Each row is a different policy, and each column an evaluation rollout from a different initial configuration. The goal is highlighted in yellow. Our method is able to achieve high success rates after 17 hours of training.

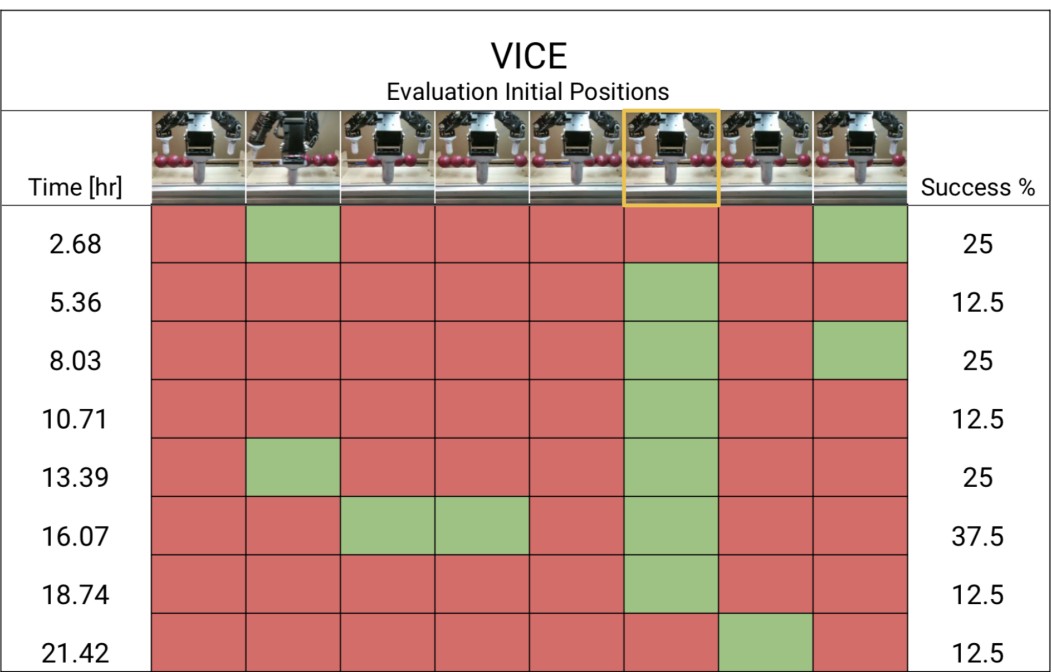

Figure 16: These are the results of evaluation rollouts on the valve rotation task in the real world using the VICE single goal baseline. The policies fail to evaluate consistently, except when the initial configuration matches the goal configuration.

