# OpenReview forum: "The Ingredients of Real World Robotic Reinforcement Learning"
_ICLR.cc/2020/Conference — Accept (Spotlight)_

### Official Review · AnonReviewer3 · 2019-10-10
**Official Blind Review #3**

**Rating:** 8

**Review:**

The paper takes seriously the question of having a robotic system learning continuously without manual reset nor state or reward engineering. The authors propose a first approach using vison-based SAC, shown visual goals and VICE, and show that it does not provide a satisfactory solution. Then they add a random pertubation controller which brings the robot or simulated system away from the goal and a VAE to encode a compressed state, and show that it works better.

The paper is a nice read, it contains useful messages thus I'm slightly in favor of accepting it, but I may easily change my mind as it suffers from serious weaknesses.

First, and most importantly, the experimental study is very short, the authors have chosen to spend much more space on careful writing of the problem they are investigating.

To mention a few experimental weaknesses, in Section 6.2 the authors could have performed much more detailed ablation studies and stress in more details the impact of using the VAE alone versus using the random pertubation controller alone, they could say more about the goals they show to the system, etc. There is some information in Figure 7, but this information is not exploited in a detailed way. Furthermore, Figure 7 is far to small, it is hard to say from the legend which system is which.

About Fig.8, we just have a qualitative description, the authors claim that without instrumenting they cannot provide a quantitative study, which I don't find convincing: you may instrument for the sake of science (to measure the value of what you are doing, even if the real-world system won't use this instrumentation).

So the authors have chosen to spend more space on the positionning than on the empirical study, which may speak in favor of sending this paper to a journal or magazine rather than a technical conference. But there is an issue about the positionning too: the authors fail to mention a huge body of literature trying to address very close or just similar questions. Namely, their concern is one the central leitmotives of Developmental Robotics and some of its "subfields", such as Lifelong learning, Open-ended learning, Continual learning etc.  The merit of the paper in this respect is to focus on a specific question and provide concrete results on this question, but this work should be positionned with respect to the broader approaches mentioned above. The authors will easily find plenty of references in these domains, I don't want to give my favorite selection here.

Knowing more about the literature mentioned above, the authors could reconsider their framework from a multitask learning perspective: instead of a random perturbation controller, the agent may learn various controllers to bring the system into various goal states (using e.g. goal-conditioned policies), and switching from goal to goal to prevent the system fro keeping stuck close to some goal.

More local points:

In the middle of page 5, it is said that the system does not learn properly just because it is stuck at the goal. This information comes late, and makes the global message weaker.

in Fig. 4, I would like to know what is the threshold for success.



**Experience Assessment:**

I have published one or two papers in this area.

**Review Assessment: Checking Correctness Of Derivations And Theory:**

N/A

**Review Assessment: Checking Correctness Of Experiments:**

I assessed the sensibility of the experiments.

**Review Assessment: Thoroughness In Paper Reading:**

I read the paper at least twice and used my best judgement in assessing the paper.

---

> ### Author Response · Authors · 2019-11-13
> **Response to Reviewer 3**
>
> We thank the reviewer for their detailed, insightful and constructive feedback! We acknowledge a number of clarity issues in the presentation and positioning of our results, which make the actual results somewhat hard to understand. We have updated the paper to make several of the points much more clear and have run additional hardware experiments to address the points raised in the review, as described in detail below:
>
> “they could say more about the goals they show to the system, etc.”
> -> We have added some visualizations about goals provided to the system to Appendix C.
>
> “you may instrument for the sake of science (to measure the value of what you are doing, even if the real-world system won't use this instrumentation).”
> -> Yes, this is a good point! We have now performed these experiments on the hardware and have included additional comparisons to baselines on hardware in Section 6.3, Fig 8. We find that the same trends observed in simulation hold on the hardware as well.
>
> “In Fig. 4, I would like to know what is the threshold for success.”
> -> Fig. 4 analyzes the sample complexity for the task of valve rotation. The experiment is considered successful when the learned policy achieves average training performance of less than 0.15 in pose distance (defined in Appendix C.1.3) across 3 seeds. Fig. 4 has now been updated to clarify this.
>
> “In Section 6.2 the authors could have performed much more detailed ablation studies and stress in more details the impact of using the VAE alone versus using the random perturbation controller alone”
> -> We have modified the Figure 7 legend and caption to make it more legible, and a discussion on the effects of the individual components based on these ablation experiments is now included in Section 6.2 in the updated manuscript. We have also updated the results after removing a small visual artifact in the environment, which allows the baselines to perform a bit better, but still maintains the same trends. We agree that the presentation of data in Figure 7 was hard to parse, and many of the comparisons (including the two requested by the reviewer) that we did actually already perform were hard to discern from the figure. The methods marked [VAE + VICE] and [RND + VICE] show the performance curves corresponding to the ablations suggested. A discussion on comparisons to explicit goal-based reset mechanisms and goal-conditioned policies has also been added to Section 6.2.
>
> “there is an issue about the positioning too: the authors fail to mention a huge body of literature trying to address very close or just similar questions...central motives of Developmental Robotics and some of its "subfields"
> -> We have expanded our related work with appropriate discussion with respect to the field of developmental robotics. The goal of our work is to enable reinforcement learning systems to handle the practicalities of learning in the real world without human instrumentation or interruption, even for a single task setting, without multi-task considerations. The insights we make should also be applicable for developmental robotics algorithms! Though our investigation doesn’t touch on all aspects of developmental robotics such as lifelong learning, open-ended learning, psychology, cognition etc., our proposed work R3L does bear strong relationship with respect to continual learning, intrinsic motivation, perceptual development, and sensory-motor development involving proprioceptive manipulation. We thank the reviewer for bringing out this interesting connection, and have added appropriate citations in the text.
>
> “authors could reconsider their framework from a multitask learning perspective...agent may learn various controllers to bring the system into various goal states and switching from goal to goal to prevent the system for keeping stuck close to some goal.”
> -> We agree that this is indeed an interesting and valuable perspective on this problem, and we have added some discussion of this to Section 6.2. We found in our experimental study that when we consider the case of using 2 goals, and switching between them (the Eysenbach et al comparison in Fig 7), it was not as effective and robust as using the perturbation controller. While this scheme chooses between only 2 goal options, and a more involved scheme could be chosen to pick multiple different goals, the performance of such an algorithm is dependent on the specific choice of goals. We find that the simpler solution via the perturbation controller can be very effective without the need for multiple meaningful alternative goals to be specified, although a better algorithm for self-supervised multi-goal selection is an interesting avenue for future work.

---

### Official Review · AnonReviewer1 · 2019-10-15
**Official Blind Review #1**

**Rating:** 8

**Review:**

This paper presents approaches to handle three aspects of real-world RL on robotics: (1)learning from raw sensory inputs (2) minimal reward design effort (3) no manual resetting. Key components:(1) learn a perturbation policy that allows the main policy to explore a wide variety of state. (2) learn a variational autoencoder to transform images to low dimensional space.

Experiments in simulation on the physical robots are performed to demonstrate the effectiveness of these components. Close related work is also used for comparison. The only concern I have is that the tasks considered involve robots that can automatically reset themselves pretty easily. I doubt that this will scale to unstable robots such as biped/quadruped, where once they fail, the recovering/resetting tasks will be as much or more difficult than the main locomotion tasks. But I understand this is too much to address in one paper and limitation is also briefly discussed in the final section.

Overall I think this is a good paper and valuable to the community.


**Experience Assessment:**

I have published one or two papers in this area.

**Review Assessment: Checking Correctness Of Derivations And Theory:**

N/A

**Review Assessment: Checking Correctness Of Experiments:**

I carefully checked the experiments.

**Review Assessment: Thoroughness In Paper Reading:**

I read the paper at least twice and used my best judgement in assessing the paper.

---

> ### Author Response · Authors · 2019-11-13
> **Response to Reviewer 1**
>
> We thank the reviewer for their encouraging feedback! We are excited about further exploring the possibilities of this line of research!

---

### Official Review · AnonReviewer2 · 2019-10-22
**Official Blind Review #2**

**Rating:** 6

**Review:**

  *Synopsis*:
  This paper focuses on current limitations of deploying RL approaches onto real world robotic systems. They focus on three main points: the need to use raw sensory data collected by the robot, the difficulty of handcrafted reward functions without external feedback, the lack of algorithms which are robust outside of episodic learning. They propose a complete system which addresses these concerns, combining approaches from the literature and novel improvements. They then provide an empirical evaluation and ablation testing of their approach and other popular systems, and show a demonstration on a real robotic system.

  Main Contributions:
  - A discussion of the current limitations of RL on real robotic systems
  - A framework for doing real world robotic RL without extra instrumentation (outside of the robot).

  *Review*:
  Overall, I think the paper is well written and provides some nice analysis of the current state of RL and robotics. I am not as familiar with the RL for robotics literature, but from some minor snooping around I believe these ideas to be novel and useful for the community. I have a few suggestions for the authors, and a few critical pieces I would like added to the main text.

  Critical additions:
  1. I would like some more details on your simulation experiments. Specifically:
    - How many runs were your experiments?
    - What are the error bars on your plots?
    - What ranges of hyper-parameters did you test for tuning?

  2. I would quite like the discussion of the real world tasks from the appendix to appear in the main text. Specifically, giving the evaluation metrics you mentioned in the appendix.

  Suggestions/Questions:

  S1: It is not clear if a VAE is the best choice for unsupervised representation learning for RL agents. Although a reasonable choice, Yashua Bengio recently released a look at several unsupervised techniques for representation learning in Atari which you may want to look at: https://arxiv.org/pdf/1906.08226.pdf.

  Q1: Did you try any of the other approaches on the real robotics system? Or was there no way to deploy these algorithms to your specific setup without instrumentation?

**Experience Assessment:**

I have read many papers in this area.

**Review Assessment: Checking Correctness Of Derivations And Theory:**

N/A

**Review Assessment: Checking Correctness Of Experiments:**

I carefully checked the experiments.

**Review Assessment: Thoroughness In Paper Reading:**

I read the paper thoroughly.

---

> ### Author Response · Authors · 2019-11-13
> **Response to Reviewer 2**
>
> We thank the reviewer for their insightful and constructive feedback! We have run additional hardware comparisons and quantitative evaluations as requested (Section 6.3) and have updated the paper according to your suggestions and comments to better discuss related work. We respond to individual concerns in detail below:
>
> “discussion of the real world tasks from the appendix to appear in the main text.”
> -> We have moved this discussion from the Appendix to Section 6.3. Additional comparisons to a VICE (Fu et al) baseline have been added for real world experiments in Section 6.3, Fig 8. We see that our algorithm is able to outperform this baseline on the real world tasks.
>
> “It is not clear if a VAE is the best choice for unsupervised representation learning for RL agents.“
> -> While a VAE works well in the domains we considered in this paper, we certainly agree that a VAE is not necessarily the optimal choice for all RL domains. We have updated Section 4.2 to reflect this explicitly, and have included references to Anand et al, Hjelm et al. and Lee et al as you pointed out as alternative methods for representation learning. We did not mean to claim that VAE’s were the only representation learning scheme that might suffice in this scenario, and many of the schemes suggested might also be effective.
>
> “I would like some more details on your simulation experiments…”
> -> We have updated Section 6 and Appendix C to include details about the experimental setup, both in simulation and in the real world. We have updated Fig 7 after removing a small visual artifact in the environment, which allows the baselines to perform a bit better, but still maintains the same trends.
> -- The plots are averaged over 5 random seeds for each method and task
> -- The (shaded) error regions correspond to the variance of the seeds for each curve
> -- Appendix B has been updated to include information on ranges of hyperparameters tuned, in addition to the optimal values used to generate the plots in figures 7 & 8
>
> “Q1: Did you try any of the other approaches on the real robotics system? Or was there no way to deploy these algorithms to your specific setup without instrumentation?”
> -> Yes we did add a new real-world comparison to the VICE baseline, as requested, in Fig 8. We have updated Section 6.3 with a comparison in the real world on the valve rotation and bead manipulation tasks. A quantitative evaluation corroborates findings from the simulated environments and shows that our method outperforms these methods in terms of sample efficiency and robustness.

---

> > ### Comment · AnonReviewer2 · 2019-11-14
> > **Thanks**
> >
> > Thank you for the response,
> >
> > I didn't mean to suggest you were claiming VAE is the best for this application. It was more a question of what else you have tried and motivations from using VAE. Again, I think it is a fine choice but I appreciate the added discussion to highlight this is just an algorithmic choice.
> >
> > I'm happy with the added experiments in the appendix, and think this makes the work more concrete. I'm a bit worried about the lack of trials in the simulated domains (5 runs is not enough see "Deep RL that Matters from Henderson https://www.aaai.org/ocs/index.php/AAAI/AAAI18/paper/viewPaper/16669). I would recommend for more improvements for publication (or a future submission) you increase the number of trials, and/or use the bootstrap method Henderson employs to make better confidence intervals.
> >
> > For Figure 8 (for valve rotation), I would make sure to run VICE for as many hours as you do for your method, or mention why they are different.
> >
> > Again, thank you for the response.

---

> > > ### Author Response · Authors · 2019-11-15
> > > **Updates**
> > >
> > > “I would recommend for more improvements for publication (or a future submission) you increase the number of trials, and/or use the bootstrap method Henderson employs to make better confidence intervals.”
> > > -> We have attempted to run more random seeds since your comment to address these concerns. Due to limited time before the end of the rebuttal period, we have been able to complete 5 additional seeds on the bead manipulation task, but additional seeds for the other two tasks have not finished. We have updated Fig 7 accordingly. We will add in the remaining seeds once they have completed running. To make it more clear that our method provides statistically significant results, we have also updated Fig 7 to show 95% bootstrap confidence intervals. These plots make it clear that the previous insights carry over to the case with confidence intervals and additional seeds as well.
> > >
> > > “run VICE for as many hours as you do for your method”
> > > -> We ran a longer run of VICE on hardware and updated the paper accordingly in Figs 8 and 14.

---

### Decision · Program_Chairs · 2019-12-19

**Decision:**

Accept (Spotlight)

**Comment:**

This is a very interesting paper which discusses practical issues and solutions around deploying RL on real physical robotic systems, specifically involving questions on the use of raw sensory data, crafting reward functions, and not having resets at the end of episodes.

Many of the issues raised in the reviews and discussion were concerned with experimental details and settings, as well as relation to different areas of related work. These were all sufficiently handled in the rebuttal, and all reviewers were in favour of acceptance.